# Estimation of Winter Wheat Tiller Number Based on Optimization of Gradient Vegetation Characteristics

**Fei Wu** [1,2,†], **Junchan Wang** [3,†], **Yuzhuang Zhou** [1,2], **Xiaoxin Song** [1,2], **Chengxin Ju** [1,2], **Chengming Sun** [1,2]
and **Tao Liu** [1,2,*]

---

1   Jiangsu Key Laboratory of Crop Genetics and Physiology, Jiangsu Key Laboratory of Crop Cultivation
    and Physiology, Agricultural College, Yangzhou University, Yangzhou 225009, China;
    mz120201187@stu.yzu.edu.cn (F.W.); mz120201192@stu.yzu.edu.cn (Y.Z.); mz120201166@yzu.edu.cn (X.S.);
    cxju@yzu.edu.cn (C.J.); cmsun@yzu.edu.cn (C.S.)
2   Jiangsu Co-Innovation Center for Modern Production Technology of Grain Crops, Yangzhou University,
    Yangzhou 225009, China
3   Lixiahe Institute of Agricultural Sciences of Jiangsu, Key Laboratory of Wheat Biology and Genetic
    Improvement for Low & Middle Yangtze Valley, Ministry of Agriculture and Rural Affairs,
    Yangzhou 225012, China; britena@163.com
*   Correspondence: tliu@yzu.edu.cn
†   These authors contributed equally to this work.

**Abstract:** Tiller are an important biological characteristic of wheat, a primary food crop. Accurate estimation of tiller number can help monitor wheat growth and is important in forecasting wheat yield. However, because of leaf cover and other factors, it is difficult to estimate tiller number and the accuracy of estimates based on vegetation indices is low. In this study, a gradual change feature was introduced to optimize traditional prediction models of wheat tiller number. Accuracy improved in optimized models, and model R2 values for three varieties of winter wheat were 0.7044, 0.7060, and 0.7357. The optimized models improved predictions of tiller number in whole wheat fields. Thus, compared with the traditional linear model, the addition of a gradual change feature greatly improved the accuracy of model predictions of wheat tiller number.

**Keywords:** winter wheat; tiller number; vegetation index; gradient feature; regression models

## 1. Introduction

Wheat is widely cultivated on a global scale as a major food crop [1], providing the main source of calories for humans [2]. Similar to most gramineous plants, wheat produces tillers, which develop from axillary buds on the mother bud [3]. The emergence, development, and survival of tillers are very important biological characteristics of wheat [4]. Wheat is highly adaptable to different environments and can self-regulate population size. Tillering can have positive or negative effects on wheat yield, but reasonable tillering is positively associated with wheat yield [5]. Tillering of wheat is affected by external factors, and the proper application of nitrogen (N) fertilizer can significantly affect tillering and promote tillering yield [6]. The number of tillers also increases with an increase in planting density [7]. The suitable application of phosphorus fertilizer also promotes wheat tillering. It is essential to understand changes in tiller development to properly manage wheat cultivation. Currently, tiller numbers are primarily determined in manual field investigations, which are costly in terms of manpower and material resources, and are also inefficient.

In recent years, wheat tiller numbers have been estimated using different techniques. Liu et al. [8] used image recognition to improve the efficiency of determining wheat populations. Tiller numbers in a wheat field were counted in pre-winter, turning green, and jointing growth stages, and canopy images of corresponding sample sections of wheat were obtained by smartphone and UAV. According to correlation analysis between canopy coverage and tiller number in the three stages, image recognition is a feasible approach to

estimate tiller number in a wheat field. However, in that study, only the single element of canopy coverage was considered, and the experimental sites were all wheat fields with striped sowing. Other factors of possible influence were not thoroughly studied, and as a result, the method has low adaptability. Wu et al. [9] randomly selected two to three points in different test plots during the critical growth period of wheat and scanned the wheat canopy with an active multispectrometer. The mean value measured in each plot was used as the spectral value, and the corresponding range was selected to determine the tiller number in each plot. The mean value was used as the tiller number in each plot to establish models to predict the wheat tiller number, which was followed by verification. Both NDVI and RVI models successfully predicted the wheat tiller number, although the NDVI model could better detect wheat growth dynamics. However, the approach needs further validation in different ecological regions and with different wheat varieties. Shan et al. [10] used cameras to take vertical photos of a wheat population at the jointing stage at a fixed height, while simultaneously measuring the total tiller number within the camera frame. In analysis of photos, a threshold value of 2g-r-b factor of a color brightness value was set to separate background from wheat, and then, a 24-bit true-color image was converted to a 256-color bitmap. LoG operator was used to detect edges and extract the number of edge pixels. After training, there was no significant difference between the BP neural network fitting effect and the measured value in estimating the total stem number of wheat. Although this method is suitable for specific varieties and growth periods, wider application requires further study. Li et al. [11] collected wheat images using ordinary cameras. Image segmentation was used to process the wheat images. Whole wheat was extracted, and the stem part of the wheat was removed by morphological processing. The two subtractions resulted in images containing stems. After edge detection, discontinuous stalks were obtained by using Hough linear transformation. Collinear segments were connected into a line segment by filling gaps, and the number of line segments detected was used to indicate the wheat tiller number. Flowers et al. [12] simultaneously measured tiller density and obtained aerial images, which were all taken on cloudless days. The data set was processed to calculate the relative tiller density of each location. A relative near-infrared and relative tiller density were used to predict tiller density, and rainbow-color aerial photos successfully predicted the wheat tiller number. However, in that study, data were from wheat fields with good management of weeds, pests and diseases, and plant nutrition, which can confuse tiller density with NIR digital counts. Phillips et al. [13] used real-time spectral reflectance sensors to collect data in the same direction in each planted row and also measured differences between vertical and parallel movements. Reflectance measurements of bare land or natural background were collected at each site, and a corrected NDVI was calculated using the measured reflectance values. A single regression equation between tiller density and modified NDVI was derived after compensating for light interference caused by clouds, shadows, and sun angle. With the equation, optical sensors could accurately predict wheat tiller density. Yuan et al. [14] designed a new ALHC algorithm based on manual measurement of tiller number in two 1-m row segments and collection of ground-based lidar measurement data. The AL algorithm recognizes gaps between wheat stems and performs clustering segmentation. When tillers are too close to one another, the hierarchical algorithm cannot distinguish tillers. Therefore, to complete automatic counting of wheat tillers, the HC algorithm calculates the number of tillers in each cluster. However, this method is greatly affected by planting density. With an increase in planting density, accuracy decreases, and leaves are also wrongly identified as tillers in the clustering step. Boyle et al. [15] obtained RGB images of plants at 0°, 45°, and 90° through NPCC and estimated wheat tiller number by using the Frangi algorithm. Scotford et al. [16] estimated the wheat tiller number by combining the coefficient of variation of normalized differential vegetation index (NDVI) with the composite vegetation index measured by ultrasonic sensor height output.

In conclusion, estimating the wheat tiller number has been investigated in many previous studies, and they have contributed greatly to the optimization of models to estimate the wheat tiller number. Estimating the wheat tiller number is different from estimating leaf area and biomass, and the same vegetation index or coverage may not be appropriate for completely different tiller states. Considering the challenges posed by the complexity of tillering in wheat, we designed a gradient vegetation feature based on estimates of coverage and vegetation indices in order to improve the accuracy of tiller number estimates. We classified the wheat coverage and vegetation index, obtained the micro-scale variation of the wheat coverage and vegetation index, and used it as a new variable to optimize the prediction model. Gradient characteristics were used to develop a high-precision model to estimate tiller number. With accurate estimates of tiller number, less fertilizer was applied in the place with a higher tiller number, and more fertilizer was applied in the places with a lower tiller number to promote the tiller number of wheat, so that field management could be improved.

## 2. Materials and Methods

### 2.1. Field Experiments

From 2019 to 2020, winter wheat field tests were conducted in Yizheng City (32°16′ N, 119°12′ E), Jiangsu Province, China (Figure 1). The climate in the area is north subtropical monsoon. The type of soil in this area is loamy soil. Initial (2019) soil nutrient analyses were as follows: 87.22 mg/kg hydrolytic nitrogen, 30.16 mg/kg available phosphorus, 121.35 mg/kg available potassium, and 2.5% organic matter. In 2020, the soil contained 121.36 mg/kg hydrolytic nitrogen, 32.53 mg/kg available phosphorus, 118.97 mg/kg available potassium, and 2.3% organic matter. The within-field variability of soil fertility had little effect on the experimental results. Experimental variables, including variety and N fertilizer and density treatments, are shown in Table 1. The weather on the measurement day was sunny, and the UAV flew for a short time from 10:00 to 11:30 to ensure sufficient and stable sunlight. The UAV was battery-powered with autonomy for about an hour and a half. The number of wheat tillers was recorded in field investigations.

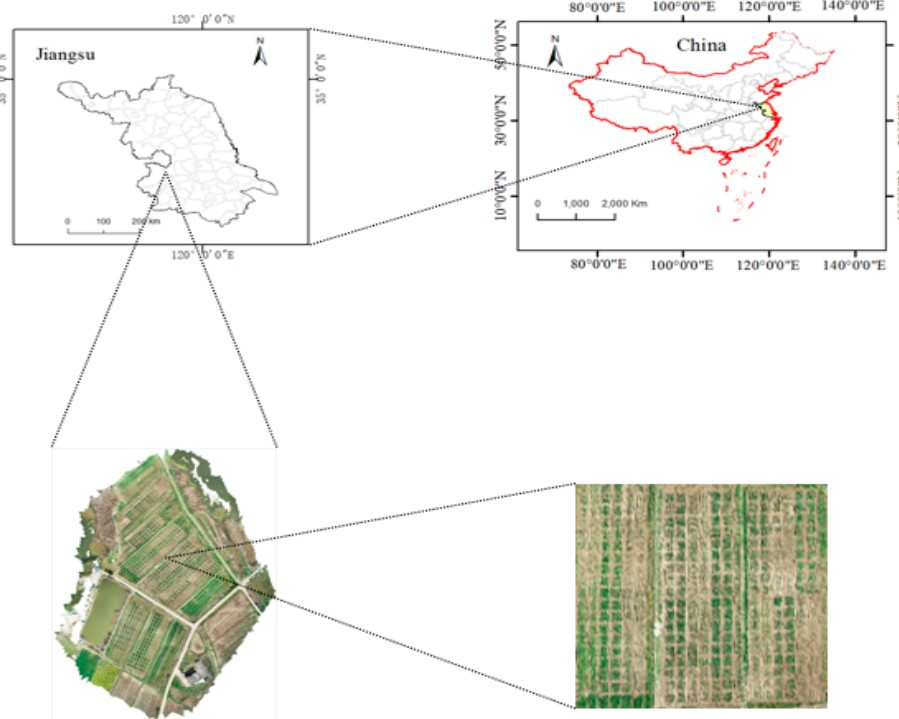

**Figure 1.** Experimental site in Yizheng City, Jiangsu Province, China.

**Table 1.** Winter wheat varieties and nitrogen fertilizer and density treatments in three different experiments (Zhenmai12: Strong gluten wheat; Yangmai16: the gluten wheat; Ningmai13: Weak gluten wheat)."Replicates" represent three Replicates of the experiment.

| Experiment | Varieties | Nitrogen Fertilizer | Density | Replicates |
|:---:|:---:|:---:|:---:|:---:|
| 1 | Zhenmai12 (P1) | 0 kg/ha (N1) | $150 \times 10^4$ ha$^{-1}$ (M1) | 3 |
| | | 125 kg/ha (N2) | $225 \times 10^4$ ha$^{-1}$ (M2) | |
| | | 225 kg/ha (N3) | $300 \times 10^4$ ha$^{-1}$ (M3) | |
| | | 375 kg/ha (N4) | | |
| 2 | Yangmai16 (P2) | 0 kg/ha (N1) | $150 \times 10^4$ ha$^{-1}$ (M1) | 3 |
| | | 125 kg/ha (N2) | $225 \times 10^4$ ha$^{-1}$ (M2) | |
| | | 225 kg/ha (N3) | $300 \times 10^4$ ha$^{-1}$ (M3) | |
| | | 375 kg/ha (N4) | | |
| 3 | Ningmai13 (P3) | 0 kg/ha (N1) | $150 \times 10^4$ ha$^{-1}$ (M1) | 3 |
| | | 125 kg/ha (N2) | $225 \times 10^4$ ha$^{-1}$ (M2) | |
| | | 225 kg/ha (N3) | $300 \times 10^4$ ha$^{-1}$ (M3) | |
| | | 375 kg/ha (N4) | | |

*2.2. Data Acquisition*

2.2.1. Multispectral UAV Image Acquisition and Processing

In the wheat overwintering period, DJI Phantom 4 multispectral spectral images were obtained. The route length was 2958 m, and there were 24 main routes. The UAV flew 20 m above the experimental site at a constant speed of 1.7 m/s with a pitch angle of 90°, and the shooting interval was 2 s. The heading overlap rate was 75%; the side overlap rate was 70%; The main technical parameters of the platform are as follows: hover accuracy: vertical: ±0.1 m, horizontal: ±0.3 m; The UAV can fly for 27 min with a pair of batteries. There are six 1/2.9-inch CMOS, including one color sensor for visible light imaging and five monochrome sensors for multispectral imaging. 2.08 million effective pixels for a single sensor (total 2.12 million pixels). The image resolutions were 21,082 pixels and 41,520 pixels. the total flight area was 1.39 ha, and 874 photos were taken. Images were obtained in the R band (600–700 nm), G band (500–600 nm), B band (350–500 nm), NIR band (800–1300 nm), and red edge band (700–760 nm). DJI Terra was used for image mosaic and multispectral image preprocessing. MATLAB was used for image processing.

2.2.2. Determination of Winter Wheat Tiller Number

The test field contained 108 plots, and two collection points were selected in each plot. Each point was marked with a 50 cm × 50 cm white box and included two rows. Wheat tiller number in the box was counted from one end. Data were recorded from a total of 216 points.

2.2.3. Feature Extraction

The ExG + Otsu method [17] was used to remove the background of the wheat field, and the ratio of wheat pixel number to total pixel number was used to represent wheat coverage (WC), which was calculated as follows:

$$WC = \frac{\text{sumArea}}{50 \text{ cm} \times 50 \text{ cm}} \tag{1}$$

where sumArea represents the number of wheat pixels and height × width represents the total pixel value.

The technical roadmap for this research is shown in Figure 2, with the following steps outlined:

(1) Data acquisition and preprocessing: a UAV obtained multispectral orthophoto images of the test site, and the images were then corrected and spliced.

(2)   Selection and structure of vegetation index: vegetation indices sensitive to tiller number were identified, the UAV multispectral range was selected to obtain images, and coverage and gradient characteristics were determined.

(3)   Analysis and comparison: vegetation indices and coverage were analyzed with wheat tiller number and results were compared after optimization of gradient special diagnosis.

(4)   Estimation of wheat tiller number in the whole field: models that provided good estimations were used to predict wheat tiller number in the whole field.

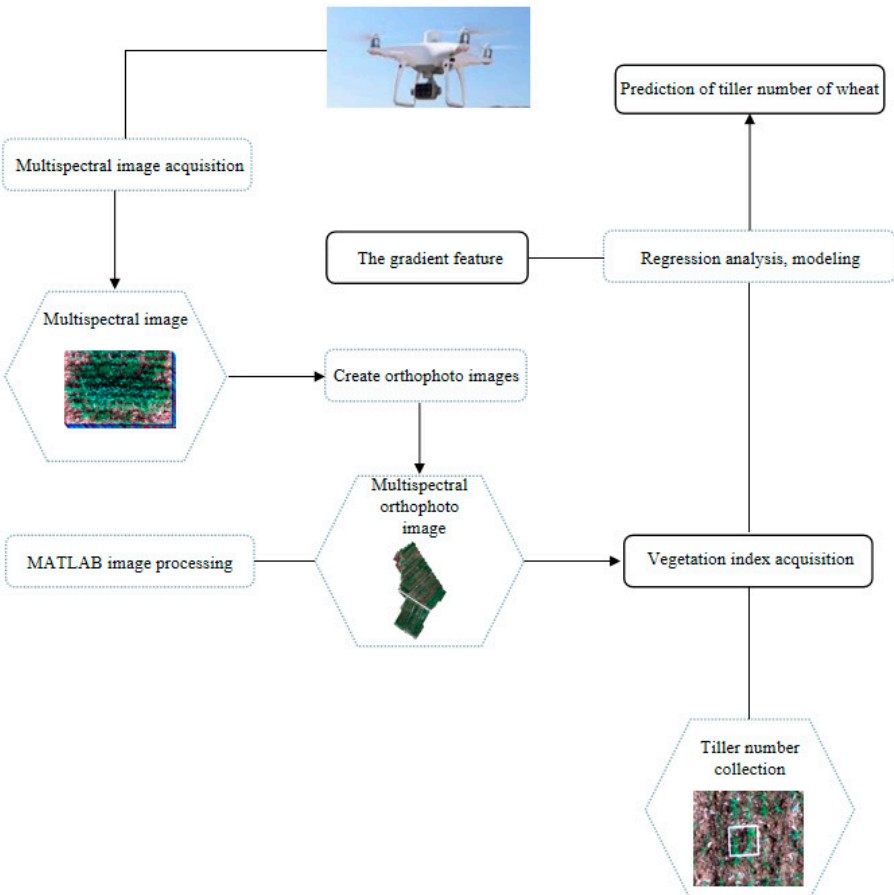

**Figure 2.** Technology Road map.

2.2.4. Analysis of Characteristics

Under the same N application level, tiller number per unit area increased with an increase in planting density. In addition, the number of tillers per unit area increased with an increase in N fertilizer at the same density. With an increase in tiller number, the coverage also increased. Therefore, it is theoretically feasible to estimate wheat tiller number based on the coverage.

Under the same N application level, with an increase in density, the overlapping rate of wheat tillers per unit area also increased and there was no significant difference in wheat coverage under different density treatments. As a result, the number of wheat tillers per unit area estimated by coverage was lower than the measured value (Figure 3). Nitrogen fertilizer has a significant effect on the leaf area of wheat, and increasing N fertilizer application at the same density also increases the wheat leaf area [18], thus there was significant difference in wheat coverage under different nitrogen treatments. As a result, the wheat tiller number estimated by coverage was higher than the measured value (Figure 4).

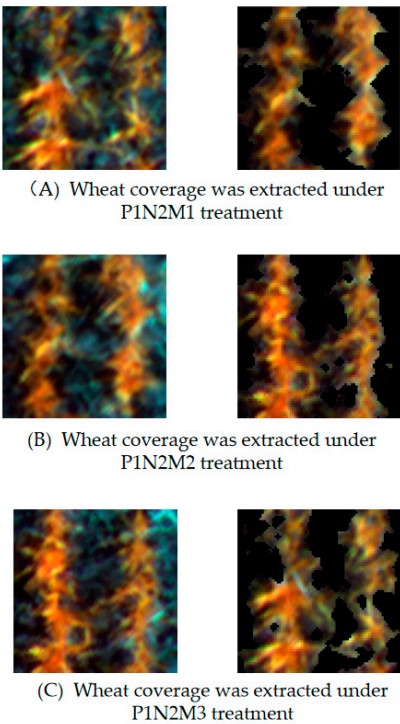

(A) Wheat coverage was extracted under P1N2M1 treatment

(B) Wheat coverage was extracted under P1N2M2 treatment

(C) Wheat coverage was extracted under P1N2M3 treatment

**Figure 3.** Wheat coverage of variety 1 (P1) in treatments with the same nitrogen fertilizer rate (N2) and different wheat densities: (**A**) M1, (**B**) M2, and (**C**) M3.

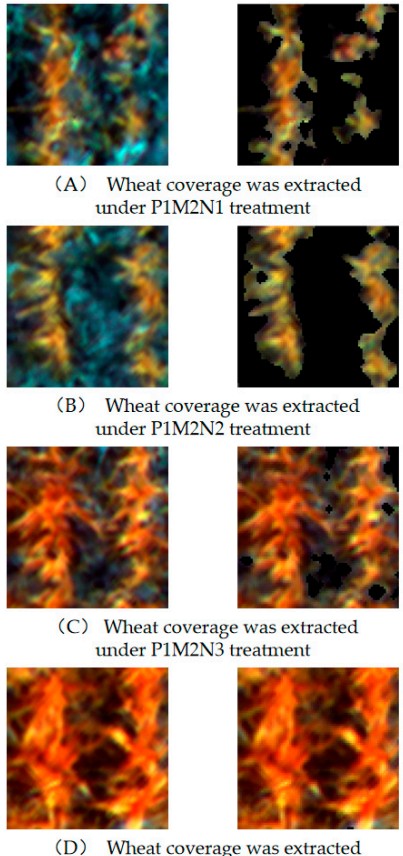

(A) Wheat coverage was extracted under P1M2N1 treatment

(B) Wheat coverage was extracted under P1M2N2 treatment

(C) Wheat coverage was extracted under P1M2N3 treatment

(D) Wheat coverage was extracted under P1M2N4 treatment

**Figure 4.** Wheat coverage of variety 1 (P1) in treatments with the same wheat densities (M2) and different nitrogen fertilizer rate wheat densities: (**A**) N1, (**B**) N2, (**C**) N3, (**D**) N4.

With NDVI, for example, a positive value indicates there is vegetation coverage, and the greater the vegetation coverage is, the greater the value [19]. In theory, it is feasible that NDVI can be used to estimate wheat tiller number. With an increase in N fertilizer application, NDVI values also increase [20]. As a result, there was no significant difference in NDVI between the high-N fertilizer and low-density treatment and the low-N fertilizer and high-density treatment (Figure 5). Therefore, NDVI is not accurate enough to predict the wheat tiller number.

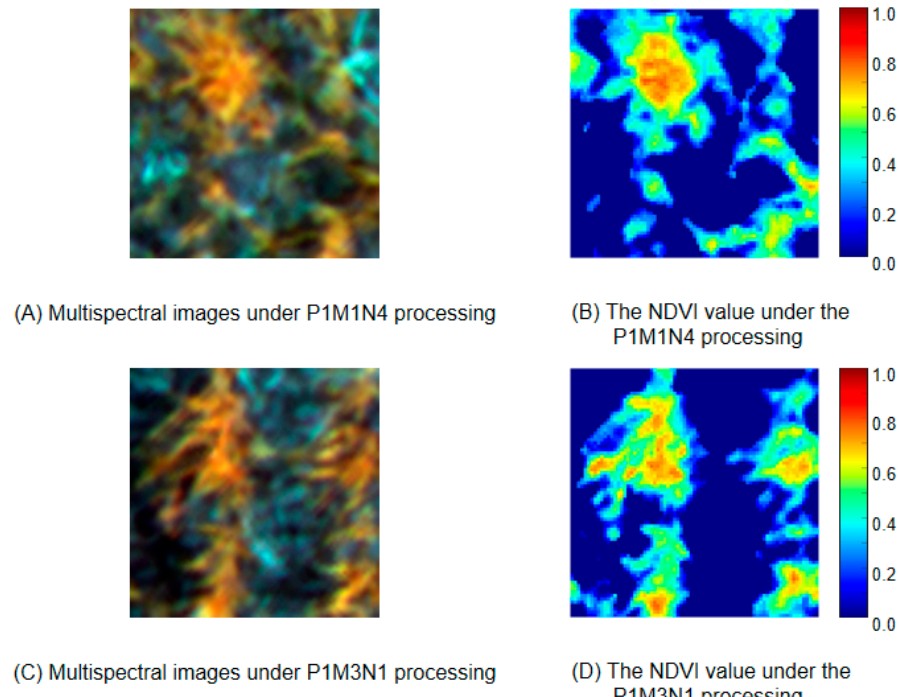

(A) Multispectral images under P1M1N4 processing

(B) The NDVI value under the P1M1N4 processing

(C) Multispectral images under P1M3N1 processing

(D) The NDVI value under the P1M3N1 processing

**Figure 5.** NDVI values for winter wheat variety 1 (P1) in (**B**) high-nitrogen fertilizer and low-density treatment (**A**,**D**) low-nitrogen fertilizer and high-density treatment (**C**).

2.2.5. Selection of Vegetation Indices

Regression analysis was used to select vegetation indices. Coverage, NDVI, and RVI were selected because of their good Regression with tiller number. The NVDI and RVI were determined as follows:

$$NDVI = (R_{NIR} - R_R)/(R_{NIR} + R_R) \tag{2}$$

$$RVI = R_{NIR}/R_R \tag{3}$$

where $R_{NIR}$ is the near-infrared reflectivity and $R_R$ is the red band reflectivity.

*2.3. Gradient Feature*

To better estimate wheat population tiller numbers, we designed a population gradient feature. When the growth process is the same, it is assumed that all tillers are evenly distributed, and the higher the population coverage is, the higher the number of tillers. However, when the number of tillers is the same but the distance between tillers is different, the more loosely arranged the tillers are, and the higher coverage (Figure 6). The degree of overlap of tiller leaves affects the vegetation index of a population. For example, the NDVI value increases with an increase in degree of overlap. However, different N contents of leaves seriously affected this relation (Figure 7). As a result, NDVI cannot accurately evaluate the degree of overlap of tiller leaves.

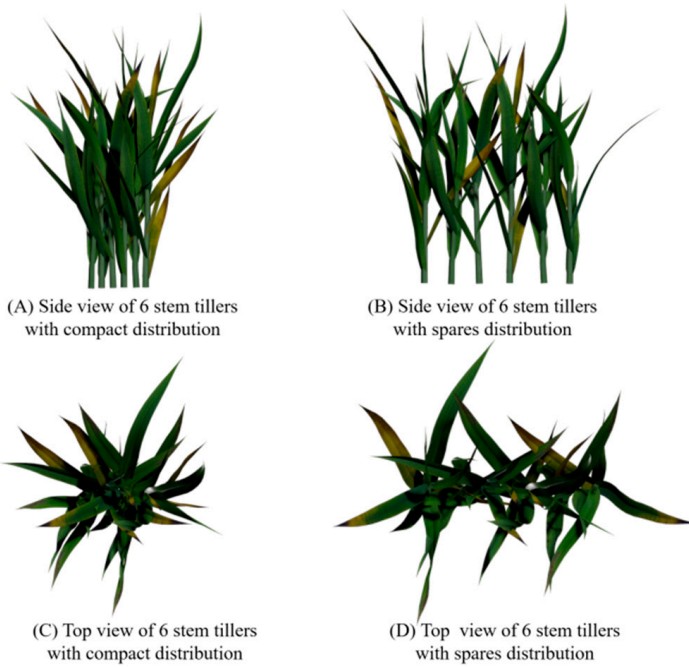

(A) Side view of 6 stem tillers
with compact distribution

(B) Side view of 6 stem tillers
with spares distribution

(C) Top view of 6 stem tillers
with compact distribution

(D) Top view of 6 stem tillers
with spares distribution

**Figure 6.** Photographs of tillering status of compact (**A**,**C**) and sparse (**B**,**D**) winter wheat.

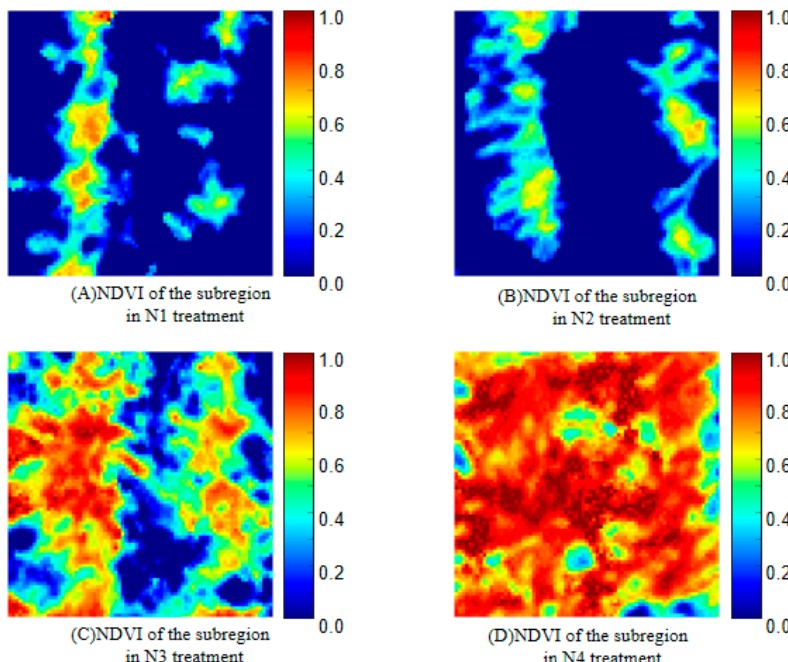

(A)NDVI of the subregion
in N1 treatment

(B)NDVI of the subregion
in N2 treatment

(C)NDVI of the subregion
in N3 treatment

(D)NDVI of the subregion
in N4 treatment

**Figure 7.** Photographs of different winter wheat tiller states: (**A**) N1, (**B**) N2, (**C**) N3 and (**D**) N4.

Images obtained by UAV were cut into regions of 50 cm × 50 cm (Figure 8A). The Otsu algorithm was used to extract wheat seedling area and calculate the vegetation index of that wheat seedling area (Figure 8B). According to the vegetation index, wheat seedling area was divided into four types: NDVI minimum Region A, sub-small Region B, large Region C, and largest Region D. The mean values (VmA, VmB, VmC, VmD) and areas (VaA, VaB, VaC, VaD) of the four regions were respectively calculated. The coefficient of variation of the vegetation index (CVI) was calculated to obtain the gradient characteristics. The classification criteria are shown in Figure 8C, and the classification effect is shown in Figure 8D.

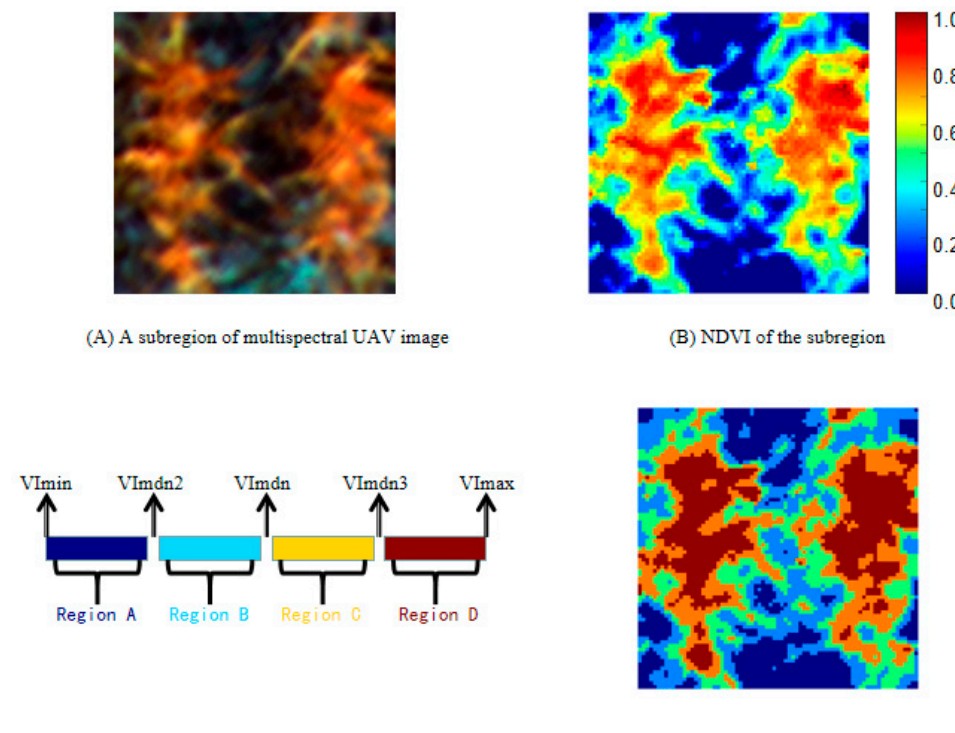

**Figure 8.** NDVI value (**B**) of wheat images (**A**) were obtained and graded (**C**) to obtain gradient feature classification map (**D**).

### 2.4. Modeling

#### 2.4.1. Linear Regression Model

Linear regression was used to analyze relations between vegetation indices and wheat tiller number, and the models established were used to predict wheat tiller number. Linear regression is a statistical method that is widely used to determine the interdependent quantitative relation between two or more variables. A regression is expressed in the form $y = W'x + e$, where the error follows a normal distribution with a mean value of 0 [21].

#### 2.4.2. Gradient Feature Optimization Model

The CVs of the mean values of the four regions of NDVI and RVI were determined (Figure 9). The CVs of the mean values of the four regions of RVI (Figure 9B) sharply increased between the larger region and the largest region. This result was observed because RVI changes significantly under high vegetation coverage, indicating that wheat tiller coverage was high in the largest region. The trend in variation among the smallest region, the smaller region, and the larger region was stable, indicating that wheat tiller coverage was low in the smallest region, the smaller region, and the larger region. The correlation between RVI and leaf area index (LAI) is high [22], and therefore, the mean value of RVI in low-coverage areas can be a good measure of leaf area. The CVs of the mean values of the four NDVI regions (Figure 9A) did not change significantly between the larger region and the largest region, because the change in NDVI was not obvious when there was high vegetation coverage.

In summary, we considered the CVs of the RVI means of the largest region and the larger region to measure the tillering coverage of the largest region and the CVs of the smallest region and the larger region to measure the tillering coverage in other areas. The RVI mean of the low-coverage area represented the LAI that was used to optimize the constructed model.

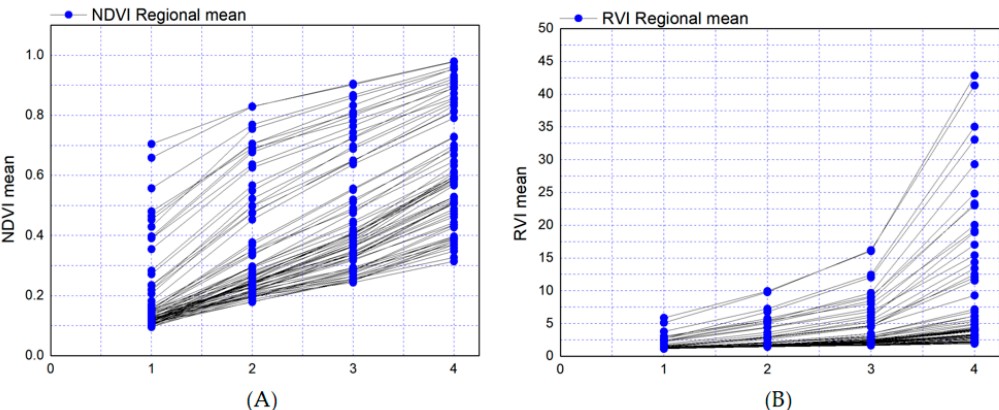

**Figure 9.** Coefficients of variation of (**A**) NDVI and (**B**) RVI among subregions.

*2.5. Statistical Analysis*

To establish models, the 216 data points were divided into three groups according to three varieties of winter wheat (Table 1). The determination coefficient ($R^2$) was used to evaluate the models, RMSE to evaluate model accuracy, and normalized root mean square error (nRMSE) to describe model accuracy. The RMSE and nRMSE were calculated as follows:

$$\text{RMSE} = \sqrt{\frac{1}{n} \sum_{i=1}^{n} \left(y_i - y_j\right)^2} \tag{4}$$

$$\text{nRMSE} = \frac{\sqrt{\frac{\sum_{i=1}^{n} \left(y_i - y_j\right)^2}{n}}}{\overline{yi}} \tag{5}$$

where $y_i$ and $y_j$ are respectively the measured value and the estimated value of the model, n is the sample size, and $\overline{yi}$ is the average value of measurement.

## 3. Results

*3.1. Estimation of Tiller Number in Wheat*

3.1.1. Unitary Linear Regression Analysis

Coverage and mean values of NDVI and RVI of the wheat regions were calculated, and measured tiller numbers of the three wheat varieties were used to construct unitary regression models (Figure 10). The trend in variation of wheat tiller number was consistent, and in all varieties, tiller number increased with increasing coverage, mean NDVI, and mean RVI. The regressions with coverage better estimated the number of tillers than those with mean values of NDVI and RVI, and the highest $R^2$ value was 0.7 for variety 3 (Figure 10C). The regressions between RVI and wheat tiller number had the lowest $R^2$ values. RVI was not suitable for tiller number estimation.

The three agronomic parameters did not adequately reflect wheat tiller number, based on the coefficients of determination for regressions between the three parameters and wheat tiller number of the three varieties. Coverage did not adequately reflect wheat tiller number because of the influence of LAI and coverage of wheat. Although NDVI could reflect LAI and coverage of wheat, it did not adequately reflect the wheat tiller number under conditions of high LAI, low LAI, and high coverage. Plant cover greatly affected RVI. When vegetation cover was high, RVI was very sensitive, but with vegetation cover <50%, sensitivity decreased significantly. Therefore, RVI also did not adequately reflect the wheat tiller number.

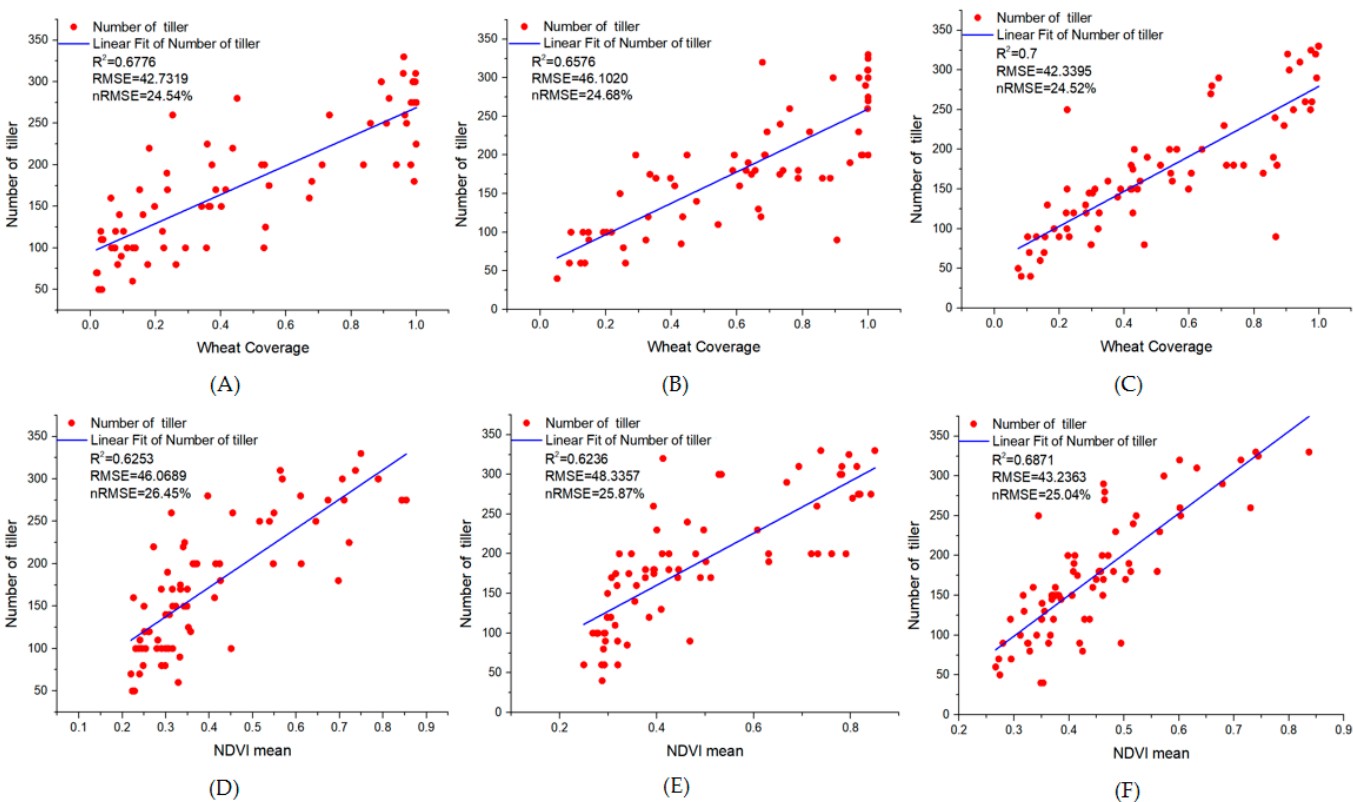

**Figure 10.** Unitary linear regression models for winter wheat variety 1 (**A**,**D**), variety 2 (**B**,**E**), and variety 3 (**C**,**F**) of number of tillers and wheat coverage (**A**,**B**), and NDVI (**D**,**E**).

### 3.1.2. Multiple Linear Regression Analysis

Multiple linear regression was used to analyze the relation between coverage and mean values of NDVI and RVI and wheat tiller number (Figure 11). When coverage, mean NDVI, and mean RVI were combined as three feature values, the coefficient of determination improved to a certain extent, compared with a single feature. The highest $R^2$ value of 0.7265 was with variety 3 (Figure 11C), which also had the lowest RMSE of 40.4228 and nRMSE of 23.41%. Compared with the mean value of RVI in predicting wheat tiller number, the $R^2$ value increased significantly, whereas compared with coverage, the $R^2$ value increased slightly. In contrast to estimating wheat tiller number by a single parameter, multiple regression with three vegetation parameters provided a more comprehensive evaluation, and thus, the $R^2$ value increased to a certaMultiple linear regression was used to analyze the relation between coverage and mean values of NDVI and RVI and wheat tiller number (Figure 11). When coverage, mean NDVI, and mean RVI were combined as three feature values, the coefficient of determination improved to a certain extent, compared with a single feature. The highest $R^2$ value of 0.7265 was with variety 3 (Figure 11C), which also had the lowest RMSE of 40.4228 and nRMSE of 23.41%. Compared with the mean value of RVI in predicting wheat tiller number, the $R^2$ value increased significantly, whereas compared with coverage, the $R^2$ value increased slightly. In contrast to estimating wheat tiller number by a single parameter, multiple regression with three vegetation parameters provided a more comprehensive evaluation, and thus, the $R^2$ value increased to a certain extent.

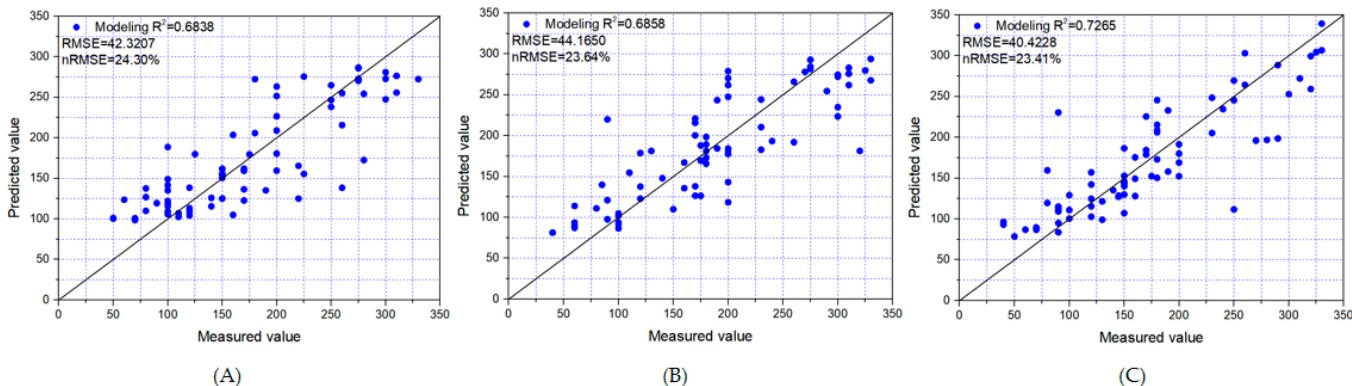

**Figure 11.** Multiple linear regression models of winter wheat tiller number with coverage, mean NDVI, and mean RVI for (**A**) variety 1, (**B**) variety 2, and (**C**) variety.

### 3.1.3. Estimation of Wheat Tiller Number after Optimization with Gradient Characteristics

In previous studies, unitary regression and multiple regression models were used to predict the wheat tiller number. Because R2 values were not very high, we attempted to improve prediction of wheat tiller number by adding gradient features (Figure 12). Compared with other prediction models, R2 values for the three varieties improved after optimization with gradient features. For variety 3, the R2 was 0.7357 and the nRMSE was 23.02% (Figure 12C), whereas for variety 1, the R2 value increased by 0.0206 (Figure 12A).

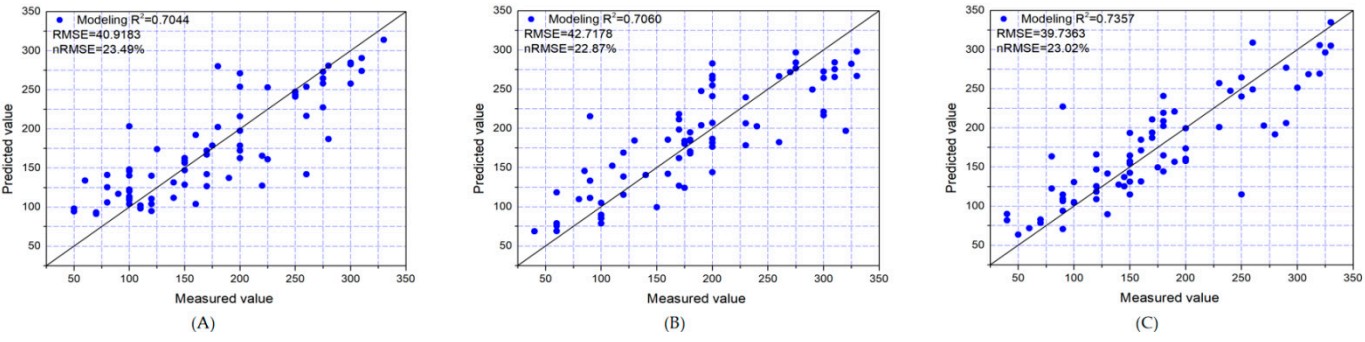

**Figure 12.** Models optimized with gradient vegetation features for winter wheat (**A**) variety 1, (**B**) variety 2, and (**C**) variety 3.

The wheat tiller number was predicted under the same N fertilizer treatment and different density treatments. For variety 1, the highest R2 value of the model to predict wheat tiller number under different density treatments was in the N3 treatment (Figure 13C), reaching 0.8953, with a RMSE of 32.3377 and nRMSE of 17.72%. For variety 2, the highest R2 value of the model to predict the wheat tiller number under different densities was in the N4 treatment (Figure 13H), reaching 0.8219, with a RMSE of 35.8180 and nRMSE of 18.29%, For variety 3, the highest R2 value of the model to predict the wheat tiller number under different densities was in the N4 treatment (Figure 13L), reaching 0.9120, with a RMSE of 28.0245 and nRMSE of 15.10%.

Wheat tiller number was also predicted under the same density treatment and different N treatments. For variety 1, the highest $R^2$ value of the model to predict wheat tiller number under different N treatments was in the M2 treatment (Figure 14B), reaching 0.8277, with a RMSE of 30.5286 and nRMSE of 16.84%. For variety 2, the highest $R^2$ value of the model to predict wheat tiller number under different N treatments was in the M1 treatment (Figure 14D), reaching 0.8925, with a RMSE of 43.0726 and nRMSE of 31.91%. For variety 3, the highest $R^2$ value of the model to predict wheat tiller number under different N treatments was in the M2 treatment (Figure 14I), reaching 0.8853, with a RMSE of 39.0222 and nRMSE of 17.70%.

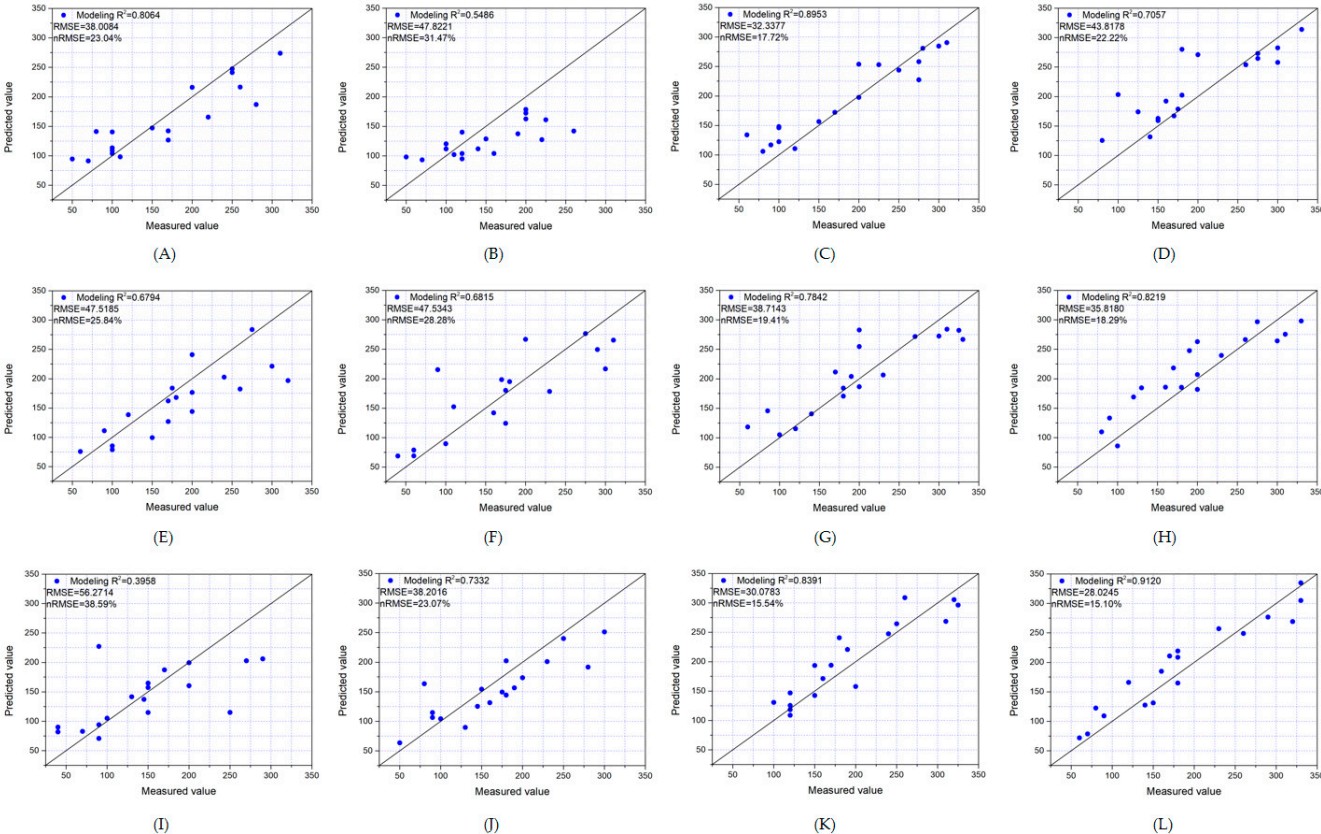

**Figure 13.** Prediction models for tiller number of different winter wheat varieties under the same nitrogen fertilizer treatment and different density treatments. (**A–D**) Variety 1; (**E–H**) variety 2; and (**I–L**) variety 3. There were four nitrogen treatments: (**A,E,I**) N1; (**B,F,J**) N2; (**C,G,K**) N3; and (**D,H,L**) N4.

### 3.2. Estimation of Wheat Tiller Number in the Whole Field

The model optimized by gradient characteristics was selected for each variety to predict the wheat tiller number in the whole field (Figure 15). For variety 1 (Figure 15A), under the same N fertilizer treatment, tiller number increased with the increase in N fertilizer application rate, and under the same N fertilizer treatment, tiller number changed little with the increase in density. For varieties 2 (Figure 15B) and 3 (Figure 15C), under the same density treatment, tiller number increased with the increase of N application rate. Under the same N treatment, in the high N treatment, the number of tillers increased with the increase in density, whereas in the low N treatment, the number of tillers remained at a generally low level. Thus, varieties 2 and 3 were greatly affected by N fertilizer. Among the three varieties, variety 1 had good and uniform tiller growth in the whole field (Figure 15A). By contrast, variety 3 had the lowest tiller number in the whole field (Figure 15C), although tiller number increased dramatically in the high-N fertilizer and high-density treatment.

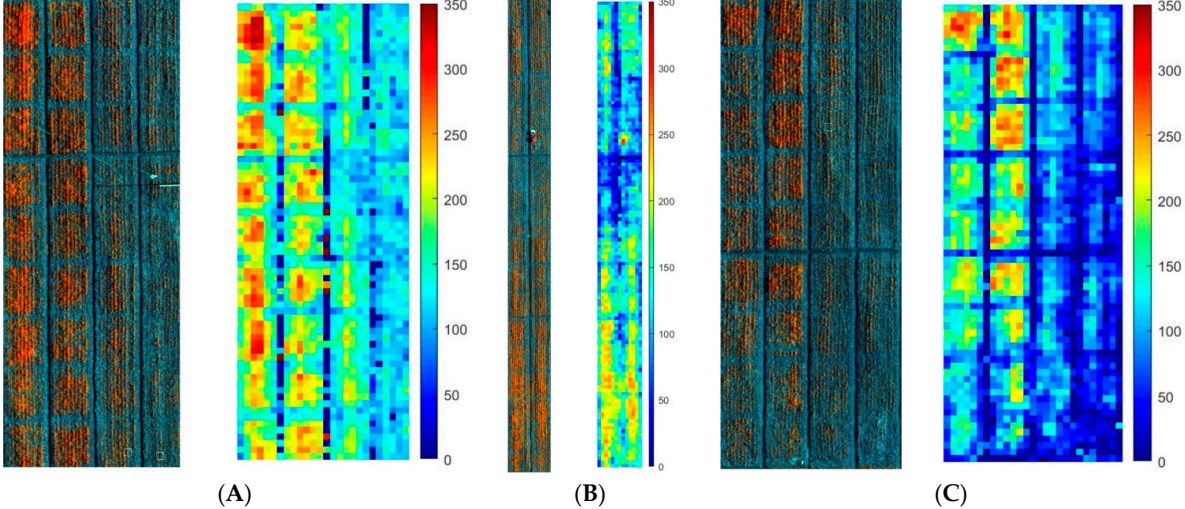

**Figure 14.** Prediction models for tiller number of different winter wheat varieties under the same density treatment and different nitrogen treatments. (**A**–**C**) Variety 1; (**D**–**F**) variety 2; and (**G**–**I**) variety 3. There were three density treatments: (**A,D,G**) M1; (**B,E,H**) M2; and (**C,F,I**) M3.

**Figure 15.** Models optimized by gradient characteristics were used to estimate winter wheat tiller numbers in the whole field. (**A**) Variety 1, (**B**) variety 2, and (**C**) variety 3.

## 4. Discussion

Tiller counts are important when monitoring the development of many plants, especially wheat [15]. Most studies that estimate plant density use ground-based non-contact measurements, focusing on relatively large plants [23]. However, in crops such as wheat, with small and variable spacing between plants with narrow leaves, leaf overlap between adjacent plants and many tillers makes visual counts difficult in the field, even when plants have more than three leaves [24]. In addition, different N treatments and planting densities significantly affect the tillering of wheat, which greatly increases the difficulty in estimating the wheat tiller number.

In this study, the tiller number estimation model of wheat was optimized and predicted by introducing gradient feature, compared with the prediction of tiller number of wheat by single elements of canopy coverage, NDVI, and RVI [8,9]. We took into account the influence of other factors and designed experimental schemes of different varieties and treatments to make the results more persuasive. Yuan et al. used LiDAR to obtain data and ALHC algorithm to successfully predict wheat tiller number, but the prediction accuracy would decrease with the increase of wheat planting density [14]. However, the prediction model optimized by gradient features could predict wheat tiller numbers well under different planting densities, showing better adaptability compared with LiDAR. Ni et al. used the penetrability of X-ray to predict the tiller number of wheat by CT. Due to the X-ray attenuation within tillers, as all tillers can be seen in the transverse section image of the wheat culms, and the tiller number can be determined through image analysis. Nevertheless, the generation of section images needs to scan the objects at hundreds of different angles, and the reconstruction has a very long computation time. So, the application of CT for real-time imaging is limited due to its low speed [25]. Compared with CT, UAV can obtain images faster and more conveniently, saving a lot of time; Boyle et al. obtained RGB images of plants through NPCC and estimated wheat tiller number by using the Frangi algorithm. This method mainly estimates the tiller number of potted wheat but does not estimate the tiller number of wheat in the whole field [15]. Our method can estimate the tiller number of wheat in the whole field, which has greater significance.

The accuracy of wheat tiller number estimates in models optimized by gradual vegetation features was improved, but the complexity of the whole process was increased. In the process of practical application, it is not easy for us to obtain prior information such as soil N content, so the specific application value of the method introduced in this paper may be weakened. In the next study, more spectral information through hyperspectral images is expected to be obtained to reduce the complexity of the optimization process. Information on soil nitrogen content will be obtained to reduce the impact of soil nitrogen content on the vegetation index, which may further improve the experimental accuracy.

## 5. Conclusions

In this study, we designed a field experiment with different varieties of winter wheat and different N fertilizer and density treatments. Multispectral images were collected by UAV, and models were established to estimate the wheat tiller number by using data on vegetation and gradient characteristics. The primary conclusions were as follows: (1) the combination of gradient and other vegetation features improved the accuracy of wheat tiller number estimates; (2) the accuracy of the wheat tiller number estimates in models optimized by gradual vegetation features was higher than that of other models, and $R^2$ values for the three varieties were 0.7044 (P1), 0.7060 (P2), and 0.7357 (P3), it could be used to effectively estimate the wheat tiller number in a whole field and more intuitively reflected the wheat tiller number in a whole field; and (3) models optimized with gradient characteristics could better estimate the wheat tiller number under different nitrogen treatments, planting densities, and growth processes.

**Author Contributions:** T.L., C.J. and C.S. conceived and designed the experiments; T.L. and F.W. performed the experiments; F.W. and T.L. analyzed the data and wrote the original manuscript; F.W., J.W., Y.Z., X.S. and T.L. reviewed and revised the manuscript. This section is not mandatory but may be added if there are patents resulting from the work reported in this manuscript. All authors have read and agreed to the published version of the manuscript.

**Funding:** This research was funded by Special Fund for Independent Innovation of Agricultural Science and Technology in Jiangsu, China (grant number CX(21)3063), National Natural Science Foundation of China (32172110, 32001465, 31872852), the Key Research and Development Program (Modern Agriculture) of Jiangsu Province (BE2020319).

**Acknowledgments:** We thank LetPub www.letpub.com (accessed on 30 November 32020) for its linguistic assistance during the preparation of this manuscript.

**Conflicts of Interest:** The authors declare no conflict of interest.

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
