# Peer review of "Estimation of Winter Wheat Tiller Number Based on Optimization of Gradient Vegetation Characteristics"

_remotesensing, doi:10.3390/rs14061338_

Round 1

Reviewer 1 Report

This paper presents the results of a field experiment that tested the effect of different varieties of fertilizer and density treatments on wheat tiller and how associated tiller numbers can be monitored with remote sensing approaches.

Overall, the paper is well written and presented. However, a number of sections contain a lot of information on wheat farming practices that are probably of more interest to the agricultural community then the remote sensing field. Yet, as they provide good context I think it is ok to leave them in as is. Nevertheless, the authors should keep in mind that the journals focus is on remote sensing and associated techniques and approaches. This means that similar detail and focus should be directed to remote sensing technical and methodological aspects. This includes the use of UAV and its survey set-up, and camera model and the choice of NDVI and RVI (e.g. why were these selected out of the wide portfolio of other indices). Little reference is made to within-field variability of soil fertility and how this might affect the results.

L114-115: provide more detail on how these field test were carried out

Table 1: Clarify what “Replicates” column stands for. Is this referring to the number of UAV flights?

Section 2.2.1 more detail needed here how the flight set-up was decided and also pixel resolution needed here.

Author Response

Reviewer #1

  1. L114-115: provide more detail on how these field test were carried out.

Response: Thank you very much for your advice. We have provided more detail on how these field test were carried out. The content is given as follow: The type of soil in this area is loamy soil. The weather on the measurement day was sunny, and the UAV flew for a short time from 10:00 to 11.30 to ensure sufficient and stable sunlight. The UAV is battery powered with autonomy for about an hour and a half.(Detailed information could be seen in Page 5 Lines 113-117).

  1. 2. Table 1: Clarify what “Replicates” column stands for. Is this referring to the number of UAV flights?

Response:"Replicates" represent three Replicates of the experiment, and it has been clarified in the manuscript. (Detailed information could be seen in Page 3 Lines 125 in the manuscript).

  1. Line 234: more detail needed here how the flight set-up was decided and also pixel resolution needed here.

Response: Thank you very much for your advice. We have provided more information about flight Settings and pixel resolution. The content is given as follow: The main technical parameters of the platform are as follows: hover accuracy: vertical: ± 0.1m, horizontal: ±0.3m; The UAV can fly for 27 minutes with a pair of batteries. There are six 1/2.9-inch CMOS, including one color sensor for visible light imaging and five monochrome sensors for multispectral imaging. 2.08 million effective pixels for single sensor (total 2.12 million pixels). The image resolutions were 21082 pixels and 41520 pixels. (Detailed information could be seen in Page 4 Lines 133-138).

Reviewer 2 Report

The authors aimed to estimate winter tiller number with UAV-based multispectral imagery and introduced a gradual change feature to optimize traditional prediction models of wheat tiller number. The improvement was significant with the new proposed method. The manuscript is clearly written and reports a valuable scientific topic. My major concern regards the novelty of the approach, which must be clearly highlighted. Moreover, some modifications are required as well as some parts need to be clarified. In the following, more detailed comments are as follows.

  • In the Equation 1, height×width is 50 cm×50 cm, that is a constant number.
  • Only use two VIs might be not so convincing, why not try more VIs since there were five bands in the imagery?
  • Figs 3, 4 and 5 were in low quality and difficult to read, and necessary legend was needed.
  • In fig10 G, H and I, linear regression was improper, and RVI was not suitable for tiller number estimation.
  • The legend of Fig 15 was blurry.
  • The discussion mainly contained the results from this study, other findings from literature should be added. This manuscript used VIs to estimate tiller number, what’s the difference of other methods, such as LiDAR.
  • Line 418 was not a complete sentence.

Author Response

Reviewer #2

  1. In the Equation 1, height×width is 50 cm×50 cm, that is a constant number.

Response: Thank you very much for your advice. We have revised this part of the content. (Detailed information could be seen in Page 4 Lines 151-152.).

  1. Line 47: Only use two VIs might be not so convincing, why not try more VIs since there were five bands in the imagery?

Response: We have verified multiple VIs through pre-experiment and found that these two have the best effect, so only these two are selected.

  1. Figs 3, 4 and 5 were in low quality and difficult to read, and necessary legend was needed.

Response: Thank you very much for your advice. We have revised this part of the content. Figure 3 and Figure 4 are renderings showing wheat images extracted under different nitrogen density conditions, so no legend is provided .

  1. ·In fig10 G, H and I, linear regression was improper, and RVI was not suitable for tiller number estimation.

Response: Thank you very much for your advice. We have deleted that part. (Detailed information could be seen in Page 11 Lines 283 and Page 12 Lines 293).

  1. 5. The legend of Fig 15 was blurry.

Response: Thank you very much for your advice. We have revised this part of the content. (Detailed information could be seen in Page 15 Lines 376.).

  1. The discussion mainly contained the results from this study, other findings from literature should be added. This manuscript used VIs to estimate tiller number, what’s the difference of other methods, such as LiDAR.

Response: Thank you very much for your advice. We have rewritten this section. The content is given as follow: In this study, the tiller number estimation model of wheat was optimized and predicted by introducing gradient feature, compared with the prediction of tiller number of wheat by single elements of canopy coverage, NDVI and RVI [8,9]. we took into account the influence of other factors and designed experimental schemes of different varieties and treatments to make the results more persuasive. Yuan et al. used LiDAR to obtain data and ALHC algorithm to successfully predict wheat tiller number, but the prediction accuracy would decrease with the increase of wheat planting density[14]. However, the prediction model optimized by gradient features could predict wheat tiller numbers well under different planting densities, showing better adaptability compared with LiDAR. Ni et al. used the penetrability of X-ray to predict the tiller number of wheat by CT. Due to the X-ray attenuation within tillers, all tillers can be seen in the transverse section image of the wheat culms, and the tiller number can be determined through image analysis. Nevertheless, the generation of section image needs to scan the objects at hundreds of different angles, and the reconstruction takes a very long computation time. So, the application of CT for real-time imaging is limited due to its low speed[25]. Compared with CT, UAV can obtain images faster and more conveniently, saving a lot of time ; Boyle et al. obtained RGB images of plants through NPCC and estimated wheat tiller number by using the Frangi algorithm. This method mainly estimates the tiller number of potted wheat but does not estimate the tiller number of wheat in the whole field[15]. Our method can estimate the tiller number of wheat in the whole field, which has greater significance.The accuracy of wheat tiller number estimates in models optimized by gradual vegetation features was improved, but the complexity of the whole process increases a lot. In the process of practical application, it is not easy for us to obtain prior information such as soil N content, so the specific application value of the method introduced in this paper may be weakened. In the next study, we hope to obtain more spectral information through hyperspectral images, to reduce the complexity of the optimization process, And we will try to measure the soil nitrogen content to reduce the impact of soil nitrogen content on vegetation index, which may further improve the experimental accuracy.(Detailed information could be seen in Page 16 Lines 389-418).

References

  1. Liu, J.; Zheng, C.; Li, Y.; Li, Z.; Fu, H.; Zhang, W. Rapid Diagnosis Technology of Wheat Stem Number Based on Canopy Image Processing. J. Henan Agric. Sci. 2019, 48, 174-80.
  2. Wu, J.; Yue, S.; Hou, P.; Meng, Q.; Cui, Z.; Li, F.; Chen, X. Monitoring Winter Wheat Population Dynamics Using an Active Crop Sensor. Spectrosc. Spect. Anal. 2011, 31, 535-8.
  3. Fang, Y.; Qiu, X.; Guo, T.; Wang, Y.; Gui, L. An automatic method for counting wheat tiller number in the field with terrestrial LiDAR. Plant Methods. 2020, 16, 132.
  4. Boyle, R.D.; Corke, F.M.K.; Doonan, J.H. Automated estimation of tiller number in wheat by ribbon detection. Mach. Vision Appl. 2016, 27, 637-46.
  5. Jiang, N.; Yang, W.; Duan, L.; Xu, X.; Huang, C.; Liu, Q. Acceleration of CT reconstruction for wheat tiller inspection based on adaptive minimum enclosing rectangle[J]. Comput. Electron . Agr. 2012, 85,123-133.

  1. 7. Line 418 was not a complete sentence.

Response: Thank you very much for your advice. This part has been deleted because we have rewritten the discussion.

Reviewer 3 Report

Tiller counts are important in monitoring the development of wheat and other crops. In the manuscript, the author designed a field experiment with different varieties of winter wheat and different N fertilizer and density treatments, and then collected the UAV image to estimate wheat tiller number by using data on vegetation and gradient characteristics. The result showed that models optimized with gradient characteristics could better estimate wheat tiller number under different nitrogen treatments, planting densities, and growth processes. The research topic itself is interesting and meaningful, and a complete experiment is carried out and a relatively clear conclusion is drawn in the manuscript. However, It is worth noting that the method in the manuscript optimizes the results, but the complexity of the whole process increases a lot, and how to apply it without prior knowledge is also a problem. Therefore, the author needs to make an in-depth discussion in the discussion part.

In addtion,the manuscript also has the following problems to be improved:

  • Lines 166-174: It is difficult to avoid that NDVI is affected by different N treatments. In this paper, the author plan to deal with it in different categories to obtain better results. But in the process of practical application, it is not easy for us to obtain prior information such as soil N content, so the specific application value of the method introduced in this paper may be weakened. It is suggested that the author discuss this problem in the discussion part.
  • Section 3: The definition of gradient feature is not clear. This concept is the core of this paper. It is suggested to define it clearly in concise language. In addition, the expression of this part needs to be strengthened logically. For example, the exactly  mean of the following sentence is puzzling: “the NDVI value increases with an increase in degree of overlap. However, different N contents of leaves seriously affected this relation”.
  • Section 2.4.2:this section is for model optimization. However, How to use the   gradient feature to optimize the model does not seem to be clearly described or showed. It is suggested to add the detail this part.
  • Generally speaking, the prediction result is slightly lower than the modeling result, but from the analysis of R2and RMSE indicators, the verification result of this paper is better than the modeling result. Why?
  • Why does Fig. 10 only use R2as the evaluation indicator, while Fig. 11 uses R2 and RMSE? In addition, why doesn't Fig. 11 adopt the form of scatter diagram like other models? If there are no special circumstances, the full text needs to be unified.
  • Discussion section: the current discussion is written more like the introduction, or has great similarities with the introduction. The discussion should combine the existing research results and deeply analyze the research results of this paper. It is suggested to rewrite the discussion section.
  • Conclusion section:at present, there are four points in the conclusion, but there are actually repetitions between them. It is suggested to further simplify and condense some meaningful points as the conclusion.
  • There are still some language defects in the manuscript, please modify it.

Author Response

Reviewer #3

  1. 1. Lines 166-174: It is difficult to avoid that NDVI is affected by different N treatments. In this paper, the author plan to deal with it in different categories to obtain better results. But in the process of practical application, it is not easy for us to obtain prior information such as soil N content, so the specific application value of the method introduced in this paper may be weakened. It is suggested that the author discuss this problem in the discussion part.

Response: Thank you very much for your advice. We have added the corresponding content in the discussion section. The content is given as follow: The accuracy of wheat tiller number estimates in models optimized by gradual vegetation features was improved, but the complexity of the whole process increases a lot. In the process of practical application, it is not easy for us to obtain prior information such as soil N content, so the specific application value of the method introduced in this paper may be weakened. In the next study, More spectral information through hyperspectral images are expected to be obtained to reduce the complexity of the optimization process, Information on soil nitrogen content will be obtained to reduce the impact of soil nitrogen content on vegetation index, which may further improve the experimental accuracy.(Detailed information could be seen in Page 17 Lines 410-418).

  1. 2. Line 191: Section 3: The definition of gradient feature is not clear. This concept is the core of this paper. It is suggested to define it clearly in concise language. In addition, the expression of this part needs to be strengthened logically. For example, the exactly  mean of the following sentence is puzzling: “the NDVI value increases with an increase in degree of overlap. However, different N contents of leaves seriously affected this relation”.

Response: Thank you very much for your advice. We have enhanced the presentation logic and defined the gradient characteristics. The content is given as follow: However, different N contents of leaves seriously affected this relation (Fig. 7), As a result, NDVI cannot accurately evaluate the degree of overlap of tiller leaves.

The coefficient of variation of the vegetation index (CVI) was calculated to obtain the gradient characteristics. (Detailed information could be seen in Page 8 Lines 212-213 and Page 9 Lines 228-230).

  1. Section 2.4.2:this section is for model optimization. However, How to use the gradient feature to optimize the model does not seem to be clearly described or showed. It is suggested to add the detail this part..

Response: Thank you very much for your advice. We have added this part of the content. The content is given as follow: In summary, we considered that the CVs of the RVI means of the largest region and the larger region to measure the tillering coverage of the largest region and the CVs of  the smallest region and the larger region to measure the tillering coverage in other areas. The RVI mean of the low-coverage area represented the LAI that was used to optimize the constructed model. (Detailed information could be seen in Page 11 Lines 256-260).

  1. Generally speaking, the prediction result is slightly lower than the modeling result, but from the analysis of R2and RMSE indicators, the verification result of this paper is better than the modeling result. Why?

Response: We selected the data of the same variety with different nitrogen density treatments for modeling and then verified the data of the same nitrogen fertilizer with different density treatments and different nitrogen fertilizer treatments respectively. It was found that some R2 was higher than the modeling, while others were lower than the modeling, but the overall prediction accuracy was quite high. This shows that the method used to predict the tiller number of wheat shows good adaptability under different treatments, and the prediction accuracy is good in some treatments, but not in some treatments.

  1. Why does Fig. 10 only use R2as the evaluation indicator, while Fig. 11 uses R2 and RMSE? In addition, why doesn't Fig. 11 adopt the form of scatter diagram like other models? If there are no special circumstances, the full text needs to be unified.

Response: Thank you very much for your advice. We have revised this part of the content.  (Detailed information could be seen in Page 12 Lines 293-294 and Page 13 Lines 319-320).

  1. Discussion section: the current discussion is written more like the introduction, or has great similarities with the introduction. The discussion should combine the existing research results and deeply analyze the research results of this paper. It is suggested to rewrite the discussion section.

Response: Thank you very much for your advice. We have rewritten this section. The content is given as follow: In this study, the tiller number estimation model of wheat was optimized and predicted by introducing gradient feature, Compared with the prediction of tiller number of wheat by single elements of canopy coverage, NDVI and RVI [8,9]. we took into account the influence of other factors and designed experimental schemes of different varieties and treatments to make the results more persuasive. Yuan et al. [14] used LiDAR to obtain data and ALHC algorithm to successfully predict wheat tiller number, but the prediction accuracy would decrease with the increase of wheat planting density. However, the prediction model optimized by gradient features could predict wheat tiller numbers well under different planting densities, showing better adaptability compared with LiDAR. Ni et al. [25 ]used the penetrability of X-ray to predict the tiller number of wheat by CT. Due to the X-ray attenuation within tillers, all tillers can be seen in the transverse section image of the wheat culms, and the tiller number can be determined through image analysis. Nevertheless, the generation of section image needs to scan the objects at hundreds of different angles, and the reconstruction takes a very long computation time. So, the application of CT for real-time imaging is limited due to its low speed. Compared with CT, UAV can obtain images faster and more conveniently, saving a lot of time; Boyle et al.[15] obtained RGB images of plants through NPCC and estimated wheat tiller number by using the Frangi algorithm. This method mainly estimates the tiller number of potted wheat but does not estimate the tiller number of wheat in the whole field. Our method can estimate the tiller number of wheat in the whole field, which has greater significance.The accuracy of wheat tiller number estimates in models optimized by gradual vegetation features was improved, but the complexity of the whole process increases a lot. In the process of practical application, it is not easy for us to obtain prior information such as soil N content, so the specific application value of the method introduced in this paper may be weakened. In the next study, we hope to obtain more spectral information through hyperspectral images, to reduce the complexity of the optimization process, And we will try to measure the soil nitrogen content to reduce the impact of soil nitrogen content on vegetation index, which may further improve the experimental accuracy.(Detailed information could be seen in Page 16 Lines 389-418).

References

  1. Liu, J.; Zheng, C.; Li, Y.; Li, Z.; Fu, H.; Zhang, W. Rapid Diagnosis Technology of Wheat Stem Number Based on Canopy Image Processing. J. Henan Agric. Sci. 2019, 48, 174-80.
  2. Wu, J.; Yue, S.; Hou, P.; Meng, Q.; Cui, Z.; Li, F.; Chen, X. Monitoring Winter Wheat Population Dynamics Using an Active Crop Sensor. Spectrosc. Spect. Anal. 2011, 31, 535-8.
  3. Fang, Y.; Qiu, X.; Guo, T.; Wang, Y.; Gui, L. An automatic method for counting wheat tiller number in the field with terrestrial LiDAR. Plant Methods. 2020, 16, 132.
  4. Boyle, R.D.; Corke, F.M.K.; Doonan, J.H. Automated estimation of tiller number in wheat by ribbon detection. Mach. Vision Appl. 2016, 27, 637-46.
  5. Jiang, N.; Yang, W.; Duan, L.; Xu, X.; Huang, C.; Liu, Q. Acceleration of CT reconstruction for wheat tiller inspection based on adaptive minimum enclosing rectangle[J]. Comput. Electron . Agr. 2012, 85,123-133.

  1. Conclusion section:at present, there are four points in the conclusion, but there are actually repetitions between them. It is suggested to further simplify and condense some meaningful points as the conclusion..

Response: Thank you very much for your advice. We have revised this part of the content. The content is given as follow: the accuracy of wheat tiller number estimates in models optimized by gradual vegetation features was higher than that of other models, and R2 values for the three varieties were 0.7044 (P1), 0.7060 (P2), and 0.7357 (P3), it could be used to effectively estimate wheat tiller number in a whole field and more intuitively reflected wheat tiller number in a whole field. (Detailed information could be seen in Page 17 Lines 425-429).

  1. There are still some language defects in the manuscript, please modify it.

Response: Thank you very much for your advice. We checked the whole paper carefully and corrected the errors.

Round 2

Reviewer 3 Report

The manuscript has been revised as required and can be published after correcting individual clerical errors (for example, the serial number (4) of the Conclusion section should be changed to (3)).

Author Response

Thank you very much for your advice. We have corrected the error (Detailed information could be seen in Page 17 Lines 436).
